# The Dissolution Mechanism of Low-Molecular-Weight Organic Acids on the Sillimanite

**DOI:** 10.3390/ma16206663

**Published:** 2023-10-12

**Authors:** Chenyang Zhang, Yaling Yu, Mingfeng Zhong, Jieyi Zhuang, Huan Yang, Shaomin Lin, Zhijie Zhang, Yunying Wu

**Affiliations:** 1School of Materials Science and Engineering, Hanshan Normal University, Chaozhou 521041, China; 2598@hstc.edu.cn (C.Z.); yaxiulingwu@163.com (Y.Y.); zjy15815026723@163.com (J.Z.); yanghuan@hstc.edu.cn (H.Y.); 2School of Materials Science and Engineering, South China University of Technology, Guangzhou 510641, China; mfzhong@scut.edu.cn; 3Chaozhou Branch of Chemistry and Chemical Engineering Guangdong Laboratory, Chaozhou 521041, China; 4Guangdong Chaoshan Institute of Higher Education and Technology, Chaozhou 521041, China; yunyingwu@hstc.edu.cn

**Keywords:** low-molecular-weight organic acids, dissolution, mechanisms, sillimanite

## Abstract

The interaction between low-molecular-weight organic acids (LMWOAs) and minerals in nature has been widely studied; however, limited research has been conducted on the dissolution mechanism of sillimanite in the presence of different organic acids. In this study, the interaction between the sillimanite sample and LMWOAs (citric acid, oxalic acid, and citric/oxalic mixture) at the same pH was investigated. The dissolution rate of Si and Al was high during the initial reaction time, then slowed down in the presence of LMWOAs. The dissolution data for Si and Al from sillimanite in the LMWOAs fit well with the first-order equation (*C*_t_ = a(1 − exp(−kt))) (R^2^ > 0.991). The dissolution process of sillimanite in the organic acids was controlled by the surface chemical reaction step. The dissolution concentration of Si in aqueous citric acid was higher than that in oxalic acid. In contrast, the dissolution concentration of Al in oxalic acid was more than that in citric acid. The maximum concentrations of Si and Al in the presence of composite organic acids were 1754 μmol/L and 3904 μmol/L. The sillimanite before and after treatment with LMWOAs were studied using X-ray diffraction (XRD) and scan electron microscopy (SEM). These results are explained by the characterization of the sillimanite. Under the single acid solution, the (210) crystal plane with a high areal density of Al in sillimanite was easily dissolved by the oxalic acid, while the (120) in sillimanite with a high areal density of Si was more easily dissolved by citric acid. In the composite organic acids, the Si-O bond and Al-O bond in sillimanite were attacked alternately, leading to the formation of some deeper corrosion pits on the surface of sillimanite. The results are of interest in the dissolution mechanisms of sillimanite in the low-molecular-weight organic acids and the environmentally friendly activation of sillimanite.

## 1. Introduction

Sillimanite is a simple chain silicate mineral among the infinite structure silicate minerals. The [SiO_4_] tetrahedron in the chain silicate mineral skeleton is linked by shared oxygen atoms and extends infinitely in a one-dimensional direction [1,2] Sillimanite (Al_2_SiO_5_) has a double-stranded structure (Figure 1), in which [SiO_4_] and [AlO_4_] alternate arrangement in the chain, sharing a cation. In the other chain, there are only [AlO_6_] octahedrons, and [AlO_6_] shares two cations with each other, and connects with the same edge. The sillimanite’s crystal structure is similar to mullite. Generally, sillimanite decomposes to mullite and silica at high temperatures, with a certain volumetric expansion [3,4]. In the industry, sillimanite is added to other products as a raw material [5,6]. The volumetric expansion effect can effectively offset the volumetric shrinkage of the product at high temperatures and improve the thermal shock resistance of the products. Sillimanite can be used as raw materials for refractories [7], synthetic mullite [8,9,10,11], metal composites [12], construction and building materials [13,14,15], and so on. Thus, sillimanite is widely used in metallurgy, ceramics, and other fields.

Excellent products require highly active sillimanite. The active sillimanite not only reduces the decomposition temperature (~1000 °C) but also aids in the combination of sillimanite with the other components. The common activation methods included mechanical activation, thermal activation, and acid activation. Mechanical activation generally uses mechanical energy to break Si-O and Al-O bonds in minerals through physical actions like collisions between particles. This increases the specific surface area and surface defects of minerals, improving their reaction ability [16,17,18,19]. Thermal activation involves pre-sintering mineral raw materials below the firing temperature, causing the minerals to change from a stable and ordered crystalline state to a long-range disordered amorphous form, enhancing the reactivity of raw materials [20,21,22,23]. However, mechanical and thermal activation requires a large amount of energy and expensive equipment.

The acid activation process has been extensively studied, with a focus on simplifying the process. Acid activation can be divided into two categories: inorganic activation and organic activation. Inorganic acids with high concentrations can dissolve cations in silicate minerals, thereby increasing their reactivity [24,25,26]. However, the concentrated inorganic acids result in high consumption and disposal costs for the waste liquid.

The application of low-molecular-weight organic acids (LMWOAs) has drawn attention. The LMWOAs effectively destroyed the Si-O bond and Al-O bond in kaolinite [27]. Strongly complexing organic acids affect the stability and promote the dissolution of clay, quartz, and feldspar to a moderate extent [28]. Organic acids and their anions may affect the dissolution rate of feldspar by affecting the formation of A1^3+^ speciation in the solution. These ions themselves can affect the dissolution rate of minerals [29]. It is assumed that LMWOAs could cause damage to the crystal structure of sillimanite, as the sillimanite comprises only Al-O and Si-O bonds. However, the effect of LMWOAs on the dissolution of sillimanite (Al_2_SiO_5_) was seldom reported. In this paper, the dissolution kinetics of the framework elements (Si and Al) in LMWOAs aqueous solution were investigated, and the crystal structure and microstructure of sillimanite in the presence of oxalic acid and citric acid were also studied.

## 2. Materials and Methods

### 2.1. Materials

Oxalic acid (C_2_H_2_O_4_·2H_2_O, AR) and citric acid (C_6_H_8_O_7_·H_2_O, AR) were from Fuchen Chemical Reagents Factory (Tianjin, China). The sillimanite was sourced from Guangdong Changlong Porcelain Company Ltd. (Meizhou, China), and its median diameter was 32.15 μm (provided by the supplier). The chemical composition of sillimanite is shown in Table 1.

### 2.2. Experimental Procedure

Experimental solutions were prepared using reagent-grade chemicals and doubly de-ionized water. Four 200 mL polyethylene bottles were filled with doubly deionized water (blank sample), 40 mmol/L oxalic acid, 40 mmol/L citric acid, and composite acids (20 mmol/L oxalic acid and 20 mmol/L citric acid) added to, respectively. The solution was then adjusted to pH 4 using 0.05 M K_2_HPO_4_ and 0.05 M HCl, and the 2 mL chloroform was added to each bottle to prevent microbial degradation of the oxalic acid and citric acid and retard microbial growth. The final volume was adjusted to 200 mL. Next, 4.0 g of the sillimanite powder was added to each of the prepared solutions. The solution was stirred at 500 rpm/min for 10 min, sealed, and placed in a biochemical incubator at 25 °C. At regular intervals, the plastic bottles were moved and stirred at 500 rpm/min for 2 min. After the reaction for periods of 24 h, 48 h, 72 h, 96 h, 120 h, 480 h, 360 h, 480 h, 600 h, and 720 h, the mixtures were stirred for 3 min at the speed of 500 rpm/min, then centrifuged and filtered through 0.45 μm nylon filters. The liquid phase was washed in triplicate and stored for later use, and the solid phase was placed in an 80 °C vacuum drying oven for 48 h to remove moisture and then stored for later use.

### 2.3. Characterization of the Sample

X-ray fluorescence (XRF, PANalytical Axios PW4400 spectrometer, Almelo, The Netherlands) was used to characterize the sillimanite [27].

For the solid characterization in this paper, the sillimanite after treatment for the longest time (720 h) was used. X-ray powder diffraction (XRD) patterns were obtained using an X’pertPro Panlytical diffractometer (Almelo, The Netherlands). Morphological analysis was conducted using a Quanta 200 scanning electron microscope (SEM) equipped with a field emission gun.

### 2.4. Characterization of the Sillimanite Dissolution Solutions

The Al concentrations were characterized by the graphite furnace atomic absorption spectrometry (contrAA700, Analytik Jena AG, Jena, Germany). The concentration of Si was obtained by the Si molybdenum blue spectrophotometric method with a visible light spectrophotometer (722N, Yoke Instrument Co., Ltd., Shanghai, China).

The dissolution data of Si and Al extracted from sillimanite were fitted to the Parabolic diffusion equation (Equation (1)), the Elovich equation (Equation (2)), and the first-order equation (Equation (3)):C_t_ = a + bt^1/2^
(1)
C_t_ = a + blnt (2)
C_t_ = a(1 − exp(−kt)) (3)
where C_t_ is the Si or Al concentration in the solution after treatment for t (hour), a and b represent the kinetics parameter, and k is the rate coefficient.

## 3. Results and Discussion

### 3.1. Dissolution Kinetics of Si and Al in Different Organic Acids

The dissolution concentration of Si and Al released from sillimanite are shown in Figure 2 and Figure 3. The dissolution curves of Si and Al were found to be similar, with an increase in dissolution concentration over time. The increase rate of the Si and Al concentrations was higher during the initial reaction time, followed by a decrease in the presence of LMWOAs. After LMWOAs treatment for 720 h, a steady state was not obtained. The dissolution concentrations of Si and Al in composite organic acids were more than that in single organic acid, with maximum values of 1754 μM and 3904 μM, respectively.

The dissolution curves of Al and Si concentrations in the presence of citric acid or oxalic acid were different. The Si concentration in citric acid was higher than that in oxalic acid, while the Al concentration in oxalic acid was more than that in citric acid. The largest concentration of both Si and Al was obtained in composite acids. It was noteworthy that the Al/Si ratio in solution with organic acids was greater than 2 (Figure 4), proving that Al was easier to dissolve than Si in sillimanite (Al_2_SiO_5_). The results implied that the oxalic acid or citric acid had preferred dissolution of Al or Si. The values of the Al/Si ratio obtained for oxalic acid were the highest, as the Al was more easily extracted from the sillimanite. In the mixture of acids, the citric acid extracted Si from the sillimanite easily, compared with oxalic acid. Thus, the Al/Si ratio obtained for composite acids was not the most, when most Al and Si ions were released for a mixture of acids simultaneously.

In the process of mineral dissolution, the leaching of cations generally undergoes the following steps: (1) adsorption of anions onto the mineral surface; (2) chelation of surface cations of the mineral, breaking the chemical bonds between the surface cations and the atomic in the crystal structure of the mineral, forming independent chelate compounds; (3) diffusion of surface chelate compounds into the liquid phase. Equations (1)–(3) were used to fit the dissolution concentration data of Si and Al extracted from sillimanite with the LMWOAs. The results are shown in Table 2 and Table 3. The dissolution kinetics of Si and Al in blank samples were difficult to determine because the three kinetic equations did not fit well with the dissolution data (R^2^ < 0.906). The dissolution concentrations of Si and Al fitted well with the first-order equation in the low-molecular-weight organic acids (R^2^ > 0.991), indicating that step (2) in the leaching process was the slow step. Therefore, the leaching process was controlled by the surface reaction step. The surface reaction control model indicated that the surface chemical reaction of minerals was the rate-limiting step of the dissolution reaction. The destruction rate of the chemical bonds during the dissolution process influenced the reaction rate. The crystal skeleton of sillimanite was composed of Si and Al (Figure 1), the energy required for breaking the Si-O bond and Al-O bond was significant. A suitable direction was needed for the effective attack of sillimanite by organic acids on the Si-O bond and Al-O bond, and the chemical bond-breaking step required a long time.

### 3.2. Changes in the Crystal Structure of Sillimanite after Treatment with Different Organic Acids

Figure 5 shows the XRD pattern of sillimanite after treatment with different organic acids. In the presence of the composite acids, the characteristic peak of sillimanite crystal was the weakest, followed by oxalic acid and citric acid. The strength of the XRD pattern diffraction peak was proportional to the completeness and symmetry of the crystal plane. The two strongest peaks of the sillimanite were (120) and (210). The destruction mechanisms of the sillimanite in the organic acids were discussed by comparing with the diffraction peak intensity of (120) and (210) (I(120) and I(210)) in the paper. The higher the I(120)/I(210) ratio, the (210) was more dissolved than (120) in sillimanite.

Table 4 shows the I(120)/I(210) of sillimanite after treatment with various organic acids. The order was as follows: oxalic acid > complex acids > citric acid. It can be observed that oxalic acid had a stronger effect on the crystal plane of (210) than citric acid, and citric acid was easier to dissolve the crystal plane of (120) than oxalic acid. After treatment with the composite acids, the ratio of I(120)/I(210) was in the middle, which indicated that both crystal planes were heavily damaged, and the preferred dissolution orientation was not clear.

The results of the XRD suggested that oxalic acid and citric acid had preferred dissolution orientation on the (210) and (120) planes, respectively. The areal density of Al in the (210) crystal plane was larger than that in (120) crystal faces (Figure 6), while the areal density of Si in the (120) plane was larger than that in (210). From the perspective of chemical composition, the proportion of Al in sillimanite crystal was twice that of Si, and oxalic acid was easier to dissolve Al than citric acid. The crystal structure of sillimanite in (210) was destroyed easily in oxalic acid. Although the concentrations of oxalic acid or citric acid used in complex acids were half of the single acid, the (120) and (210) surfaces dissolved the most under the action of composite acids; moreover, the dissolution concentration of Si and Al was the highest (Figure 2 and Figure 3), which indicated that the presence of two organic acids working synergistically could efficiently dissolve Si and Al in sillimanite.

### 3.3. Dissolution Mechanism of Sillimanite in LMWOAs

The morphology of sillimanite before and after treatment with the composite acids is shown in Figure 7. The sillimanite surface appeared non-porous (Figure 7b), indicating the rapid migration of organic acid anions to the surface of sillimanite. After a reaction time of 720 h, nano-etch pits were observed on the surface (Figure 7d). These observations support the surface reaction control model for the dissolution of sillimanite and are consistent with the analysis of dissolution curves present in Table 2 and Table 3.

The dissolution mechanism of composite acids on sillimanite minerals can be described as follows: (1) oxalic acid and citric acid in solution approached the Si and Al reaction sites of sillimanite (Figure 8a). (2) The weak Al-O bond in the crystal structure of [AlO_6_] and [AlO_4_] made them susceptible to attack by the organic acids. Citric acid was more likely to attack Si-O bonds than oxalic acid (Figure 8b). (3) As the reaction progresses, Si and Al diffuse from the mineral surface to the liquid phase in a composite acid solution. The Al-rich crystal surface of (210) was more susceptible to oxalic acid attack, while the Si-rich crystal plane of (120) was more susceptible to citric acid attack. Through the synergistic effect of the two acids, citric acid, and oxalic acid alternately attacked the Si-O bond and Al-O bond, resulting in the effective dissolution of minerals (Figure 8c). (4) With increasing time, triangular-shaped corrosion pits emerged on the surface of sillimanite treated by composite organic acids, and even some small fragments of sillimanite detached from the edge of the mineral (Figure 8d). This indicates that the dissolution process of sillimanite by composite acid was controlled by the surface chemical reaction step.

## 4. Conclusions

The dissolution behavior of sillimanite in the presence of low-molecular-weight organic acids was studied by analyzing the solid and liquid phases. The rate of dissolution was high initially and then reduced in the presence of low-molecular-weight organic acids. The dissolution data of Si and Al in sillimanite were well-described by the first-order equation after treatment with organic acids (R^2^ > 0.991). The surface chemical reaction step controlled the sillimanite dissolution process in organic acids.

The maximum concentration of Si and Al, reaching values of 1754 μM and 3904 μM, respectively, was observed in the presence of composite organic acids (oxalic acid and citric acid), compared to using oxalic acid or citric acid alone. In single acid solutions, oxalic acid dissolved Al more easily, while citric acid preferentially attacked the Si site. The analysis of the solid phase suggested that the oxalic acid had a stronger effect on the crystal plane (210) in sillimanite with a high areal density of Al, while the (120) in sillimanite with a high areal density of Si was easier to be dissolved by citric acid. In the composite organic acids solution, the synergistic effect of oxalic acid and citric acid was evident, and the Si-O bond and Al-O bond were attacked alternately, enhancing the release of Si and Al in sillimanite. The dissolution of a large amount of Si and Al was attributed to the jagged edges of sillimanite and the formation of deeper triangle corrosion pits. The adsorption process of low-molecular-weight organic acids on the sillimanite surface and the break mechanism of Si-O and Al-O bonds in sillimanite is unclear and needs further study. These results are interesting in terms of understanding the dissolution mechanisms of sillimanite in LMWOAs and producing environmentally friendly high-activity sillimanite powder.

## Figures and Tables

**Figure 1 materials-16-06663-f001:**
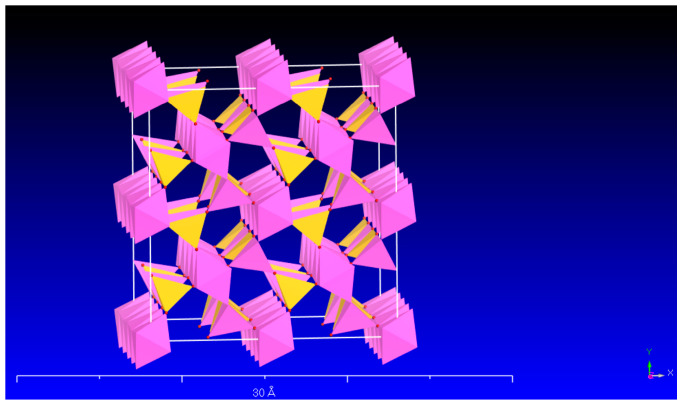
The crystal structure of sillimanite (yellow represents the [SiO_4_] and pink represents [AlO_6_]).

**Figure 2 materials-16-06663-f002:**
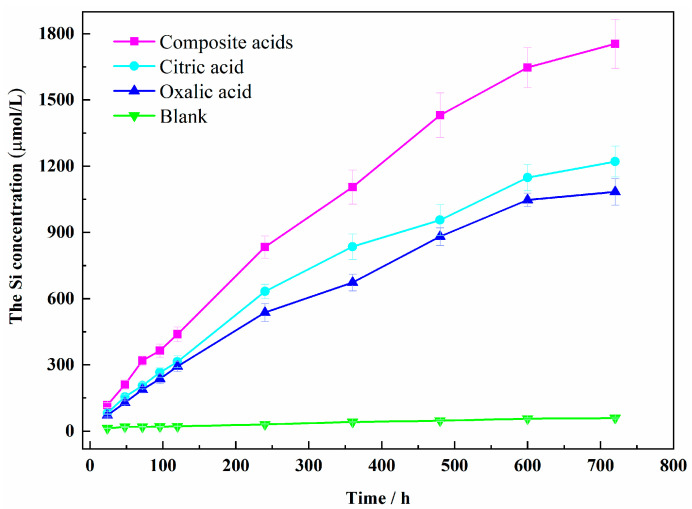
The Si concentration of different organic acids.

**Figure 3 materials-16-06663-f003:**
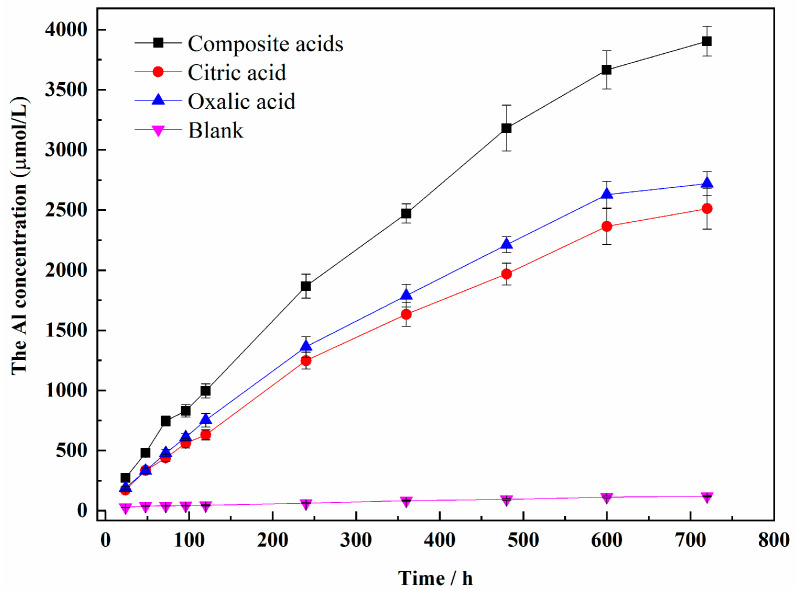
The Al concentration of different organic acids (The black line in the triangle is error bar).

**Figure 4 materials-16-06663-f004:**
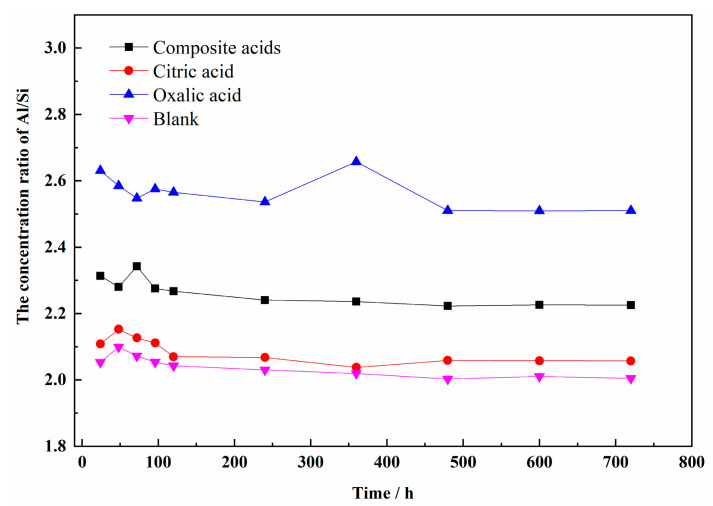
The concentration ratio of Al/Si release from sillimanite in different organic acids.

**Figure 5 materials-16-06663-f005:**
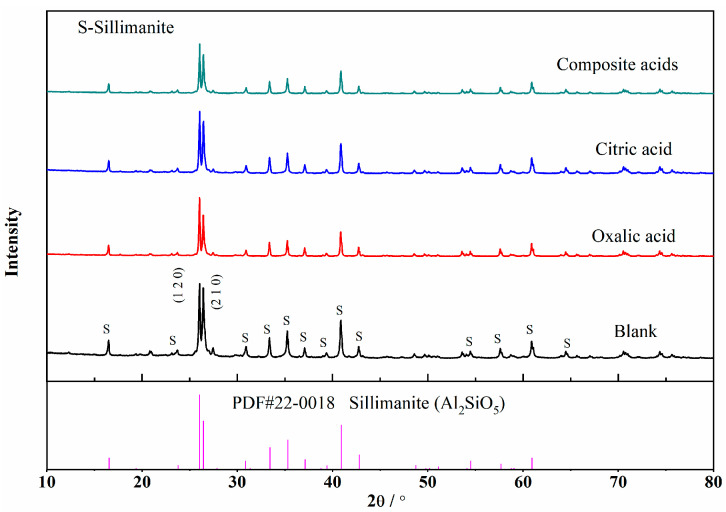
The XRD pattern of sillimanite before and after treatment.

**Figure 6 materials-16-06663-f006:**
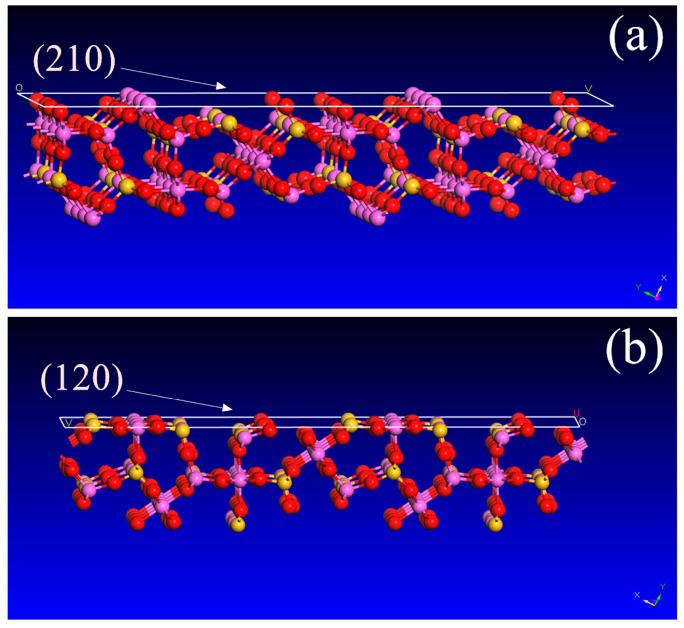
Side view of (**a**) (210) and (**b**) (120) in sillimanite (the red, yellow, and purple balls represent O, Si, and Al, respectively).

**Figure 7 materials-16-06663-f007:**
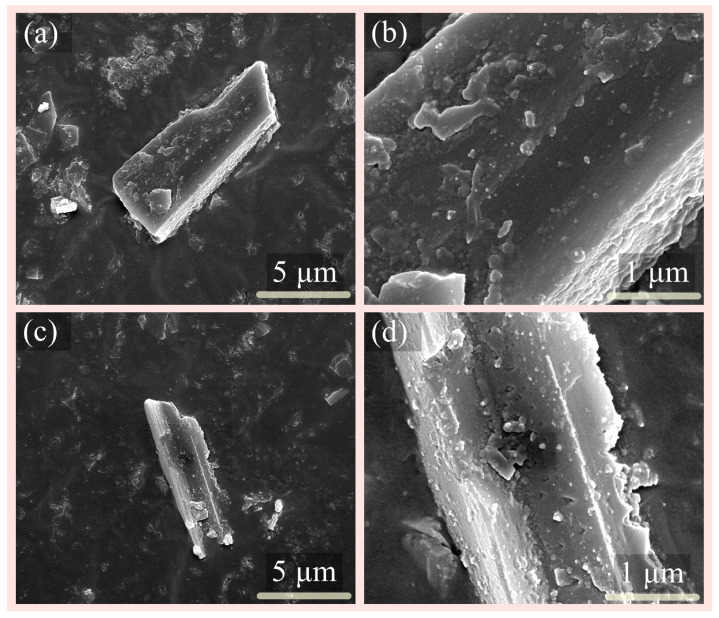
The SEM images of the sillimanite (**a**,**b**) before and (**c**,**d**) after treatment with the composite acids for 720 h.

**Figure 8 materials-16-06663-f008:**
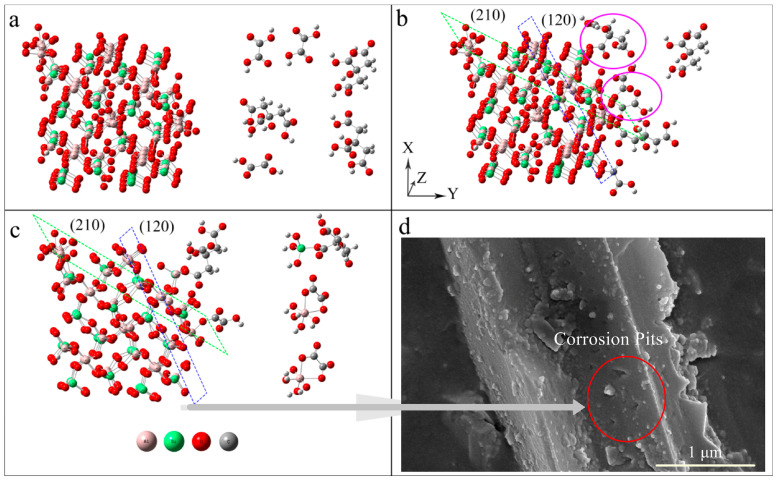
The schematic diagram of the dissolution mechanisms of sillimanite in the presence of composite organic acids. (**a**) oxalic acid and citric acid in solution (right in the image) approached the Si and Al reaction sites of sillimanite (left in the image); (**b**) The Al-O and Si-O bonds was attacked by the organic acids;. (**c**) Si and Al diffuse from the mineral surface to the liquid phase in a composite acid solution; (**d**) triangular-shaped corrosion pits emerged on the surface of sillimanite, and even some small fragments of sillimanite detached after composite acid treatment.

**Table 1 materials-16-06663-t001:** Chemical composition (wt%) of sillimanite.

Sample	SiO_2_	Al_2_O_3_	CaO	MgO	Fe_2_O_3_	Na_2_O	K_2_O	TiO_2_	P_2_O_5_	MnO	LOI ^1^
Sillimanite	41.25	55.07	0.66	0.71	0.23	0.12	0.43	0.55	0.03	0.02	0.93

LOI ^1^ = Loss on ignition.

**Table 2 materials-16-06663-t002:** The kinetic parameters of Si release from sillimanite in the different organic acids.

The Solution	$C_{t}=a+bt^{1/2}$		$C_{t}=a+blnt$		$C_{t}=a\left(1-exp\left(-kt\right)\right)$	
	b	R^2^	b	R^2^	b	R^2^
Blank	2.13	0.906	13.76	0.891	4.09 × 10^−3^	0.896
Oxalic acid	49.43	0.991	3.21 × 10^2^	0.915	1.70 × 10^−3^	0.996
Citric acid	55.29	0.993	3.61 × 10^2^	0.925	1.84 × 10^−3^	0.998
Composite acids	79.51	0.991	5.15 × 10^2^	0.911	1.93 × 10^−3^	0.998

**Table 3 materials-16-06663-t003:** The kinetic parameters of Al release from sillimanite in the different organic acids.

The Solution	$C_{t}\;=\;a\;+\;bt^{1/2}$		$C_{t}\;=\;a\;+\;blnt$		$C_{t}\;=\;a\left(1-exp\left(-kt\right)\right)$	
	b	R^2^	b	R^2^	b	R^2^
Blank	4.26	0.902	27.44	0.884	4.21 × 10^−3^	0.882
Oxalic acid	1.24 × 10^2^	0.993	8.09 × 10^2^	0.922	1.86 × 10^−3^	0.998
Citric acid	1.22 × 10^2^	0.993	7.32 × 10^2^	0.918	1.78 × 10^−3^	0.998
Composite acids	1.75 × 10^2^	0.992	1.14 × 10^3^	0.913	1.97 × 10^−3^	0.998

**Table 4 materials-16-06663-t004:** The value of I(120)/I(210) of sillimanite after treatment in the different organic acids.

Samples	Blank	Oxalic Acid	Citric Acid	Composite Acids
I(120)/I(210)	1.07	1.43	1.19	1.28

## Data Availability

Data sharing not applicable. No new data were created or analyzed in this study. Data sharing is not applicable to this article.

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
