# Peer review of "The Dissolution Mechanism of Low-Molecular-Weight Organic Acids on the Sillimanite"

_materials, 2023, doi:10.3390/ma16206663_

Round 1

Reviewer 1 Report

The manuscript has a lack of novelty.

The english in the manuscript is enough for science-journals.

Author Response

Comment 1:

The manuscript has a lack of novelty.

Our response:

We thank the referee very much for the comments.

Low molecular weight organic acids (LMWOAs) play an important role in the dissolution and structure alteration of aluminosilicate mineral, but the mechanisms of the rapid dissolution of sillimanite in the presence of different LMWOAs have been less investigated. The composites LMWOAs is the common existent forms in nature, but most research pay attention to the dissolution mechanisms of minerals in single LMWOA. Here, the dissolution kinetics of the sillimanite sample in aqueous citric acid, oxalic acid and citric/oxalic mixture were examined. In the composite organic acids' solution, the Si-O bond and Al-O bond in sillimanite were attacked alternately, leading to the formation of some deeper corrosion pits on the surface of sillimanite. The results are interesting in terms of understanding the dissolution mechanisms of sillimanite in the organic acids and the acid activation process to produce environmentally friendly high-activity sillimanite.

Reviewer 2 Report

The manuscript "The dissolution mechanism of low molecular weight organic acids on the sillimanite" by Chen-Yang Zhang , Ya-Ling Yu , Ming-Feng Zhong , Jie-Yi Zhuang , Huan Yang , Shao-Min Lin * , Zhi-Jie Zhang * , Yun-Ying Wu is acceptable. There are some issues needed to be amended, revised, and fixed prior acceptance. The comments and suggestions are as stated below. 

Introduction:

      i.         Please elaborate more on the background of study, improvement of this research as compared to previous research.

     ii.         Please change the personal pronouns such as ‘our team’ in the text

Materials and Methods:

      i.         XRF and AAS are mentioned in the manuscript. However, results have yet to be discussed in the manuscript and abstract section. Please include the information in both sections; otherwise, the authors should eliminate it from the material section. 

Results:

      i.         The discussion parts on this section are very shallow. The authors should include their critical opinion and related previous study/reference on state of the art in the manuscript.

     ii.         The characterization section is not well discussed. The authors only stated/mentioned the results obtained. The authors should discuss based on the previous studies/research (with example) that has been conducted using the specific instrument for characterization.

   iii.         Please improve the resolutions of all figures.

   iv.         The XRD results is not clear. Please revise it (Figure 5 – pdf# 38-0471—sillimanite—Al2SiO5). 

Technical Issues:

      i.         1 Loss on ignition should be written as LOI1 = Loss on ignition

     ii.         3.3. FormTable 3 – Please eliminate 3.3 From

Conclusion: Describe the study's limitations and future improvement in applying this type of material. 

References: Please cite recent articles to support the review articles as up-to-date research. 

Language Style: The English language quality is moderate and needs proofreading to meet the standard necessary for a scientific article.

Moderate editing of English language required.

Author Response

Response to the referee 2 

(comments in black and responses in blue and red)

Comment 1:

Introduction:

  1. Please elaborate more on the background of study, improvement of this research as compared to previous research.

Our response:

We thank the referee very much for the comments.

The background of the study have been elaborated.

The original text is revised as follows:

“The LMWOAs effectively destroyed the Si-O bond and Al-O bond in kaolinte [27]. Strongly complexing organic acids affect the stability and promote the dissolution of clay, quartz and feldspar to a moderate extent[28]. Organic acids and their anions may affect the feldspar dissolution rates mainly by affecting the speciation in solution of ions such as A13+ that themselves affect the mineral dissolution rate and even to form complexes at the mineral surface[29]. It is assumed that LMWOAs could cause damage to the crystal structure of sillimanite, as the sillimanite comprises only of Al-O and Si-O bonds.”

Comment 2:

Introduction:

  1. Please change the personal pronouns such as ‘our team’ in the text

Our response:

Thanks a lot for the valuable suggestions. The personal pronouns have been changed in the text. The original text is revised as follows:

“The LMWOAs effectively destroyed the Si-O bond and Al-O bond in kaolinte [27]. It is assumed that LMWOAs could cause damage to the crystal structure of sillimanite, as the sillimanite comprises only of Al-O and Si-O bonds.”

Comment 3:

Materials and Methods:

  1. XRF and AAS are mentioned in the manuscript. However, results have yet to be discussed in the manuscript and abstract section. Please include the information in both sections; otherwise, the authors should eliminate it from the material section. 

Our response:

We thank the referee very much for the comments.

The result of XRF was seldom discussed. However, the chemical composition (wt%) of sillimanite in Table 1 was characterized by XRF in the section of 2.1 Materials.

The Al concentrations in the solution was measured by the AAS, and the results were discussed in the section of 3. Results and discussion.

Comment 4:

Results:

  1. The discussion parts on this section are very shallow. The authors should include their critical opinion and related previous study/reference on state of the art in the manuscript.

Our response:

Thanks a lot for the reviewer’s valuable comments. The original manuscript is revised as follows:

“The results of the XRD suggested that oxalic acid and citric acid had preferred dissolution orientation on the (210) and (120) planes, respectively. ”

Comment 5:

Results:

  1. The characterization section is not well discussed. The authors only stated/mentioned the results obtained. The authors should discuss based on the previous studies/research (with example) that has been conducted using the specific instrument for characterization.

Our response:

Thanks a lot for the reviewer’s valuable comments. The original manuscript is revised as follows:

“.....This indicates that the dissolution of Si and Al in sillimanite is controlled by the surface reaction process in the presence of LMWOAs, as the chemical reaction process is the control step. The surface reaction control model indicated that the surface chemical reaction of minerals was the rate-limiting step of the dissolution reaction. The destruction rate of the chemical bonds during the dissolution process influenced the reaction rate. The crystal skeleton of sillimanite was composed of Si and Al (Figure 1), the energy required for breaking Si-O bond and Al-O bond was significant. A suitable direction was needed for the effective attack of sillimanite by organic acids on Si-O bond and Al-O bond and the chemical bond breaking step required a long time.”

Comment 6:

Results:

iii. Please improve the resolutions of all figures.

Our response:

Thanks a lot for the reviewer’s valuable comments. 

We have improved the resolutions of all figures to 300 dpi.

Comment 7:

Results:

  1. The XRD results is not clear. Please revise it (Figure 5 – pdf# 38-0471—sillimanite—Al2SiO5).

Our response:

We thank the referee very much for the comment.

The PDF#22-0018 was used to replace PDF#38-0471 in the manuscript.

Comment 8:

Technical Issues:

  1. 1 Loss on ignition should be written as LOI1= Loss on ignition.

Our response:

Thanks a lot for the reviewer’s valuable comments. 

The original manuscript changes as follows:

“LOI1 = Loss on ignition”

Comment 9:

Technical Issues:

  1. 3.3. FormTable 3 – Please eliminate 3.3 From.

Our response:

Thanks a lot for the reviewer’s valuable comments. The “3.3. Form” have been eliminated.

Comment 10:

Conclusion:

Describe the study's limitations and future improvement in applying this type of material.

Our response:

Thanks a lot for the reviewer’s valuable comments. The Conclusion of the paper have been revised carefully.

The original manuscript is revised as follows::

“The adsorption process of low molecular weight organic acids on the sillimanite surface and the break mechanism of Si-O and Al-O bonds in sillimaite is unclear and needs further study. These results are interesting in terms of understanding the dissolution mechanisms of sillimanite in different organic acids and the acid activation process to produce environmentally friendly high-activity sillimanite powder.”

Comment 11:

References:

Please cite recent articles to support the review articles as up-to-date research.

Our response:

Thanks a lot for the reviewer’s valuable comments. Recent articles to support the review articles have been cited in the manuscript.

The original manuscript changes as follows:

“[28]Potysz, A.; Bartz, W. Bioweathering of minerals and dissolution assessment by experimental simulations—Implications for sandstone rocks: A review. Constr. and Build. Mater. 2022, 316, 125862.

[29]Yuan, G.; Cao, Y.; Schulz, H.; Hao, F.; Gluyas, J.; Liu, K.; Yang, T.; Wang, Y.; Xi, K.; Li, F. A review of feldspar alteration and its geological significance in sedimentary basins: From shallow aquifers to deep hydrocarbon reservoirs. Earth.-Sci. Rev. 2019, 191, 114-140.”

Comment 12:

Language Style: The English language quality is moderate and needs proofreading to meet the standard necessary for a scientific article.

Our response:

Thanks a lot for the reviewer’s valuable comments. The English language in ä»–in the manuscript have been carefully checked.

The original manuscript changes as follows:

“It is noteworthy that the Al/Si ratio in solution with organic acids is greater than 2 (Figure 4), proving that Al was easier to dissolve than Si.”

“The result indicated that the dissolution of Si and Al in sillimanite was controlled by the surface reaction process in the presence of LMWOAs.”

“The result indicated that the dissolution of Si and Al in sillimanite was controlled by the surface reaction process in the presence of LMWOAs.”

“ The higher the I(120)/I(210) ratio, the (210) was more dissolved than (120) in sillimanite.”

.....

………………………………….end of response to the comments……………………………

Reviewer 3 Report

Comments and suggestions for Authors are included in the attached file.

Some parts of the text need moderate English corrections. In particular, lines 40-44 and 52-54 are difficult to follow. There is also an inappropriate use of vocabulary, e.g. in lines 192-193 the Authors write about "outstanding solubility," while the results obtained do not confirm this.

Author Response

Response to the referee 3 

(comments in black and responses in blue and red)

Comment 1:

  1. The Authors should more clearly specify the purpose of their research and clearly indicate in the manuscript what distinguishes their research from those already conducted (not just by their own research group).

Our response:

We thank the referee very much for the comments.

We have checked the Introduction carefully.

The original text is revised as follows:

“The LMWOAs effectively destroyed the Si-O bond and Al-O bond in kaolinite [27]. Strongly complexing organic acids affect the stability and promote the dissolution of clay, quartz and feldspar to a moderate extent[28]. Organic acids and their anions may affect the feldspar dissolution rate mainly by affecting the speciation in solution of ions such as A13+ that themselves affect the mineral dissolution rate and even to form complexes at the mineral surface[29]. It is assumed that LMWOAs could cause damage to the crystal structure of sillimanite, as the sillimanite comprises only of Al-O and Si-O bonds. However, the effect of LMWOAs on the dissolution of sillimanite (Al2SiO5) was seldom reported. In this paper, the dissolution kinetics of the framework elements (Si and Al) in LMWOAs aqueous solution were investigated, and the crystal structure and microstructure of sillimanite in the presence of oxalic acid and citric acid were also studied.”

Comment 2:

Materials and Methods section should be edited more thoroughly, including:

  1. a) It has been stated that the average particle size of the sillimanite used is 32.15 µm - please clarify whether the Authors evaluated this parameter themselves (if so, how) or is it just based on the data provided by the supplier.
  2. b) The Authors should explain for what purpose they added chloroform to the samples .
  3. c) The wording “After the reaction was complete”(lines 97-98) is unclear – how did the Authors evaluate this completion of reaction?
  4. d) r/min should be replaced by rpm/min.

Our response:

Thanks a lot for the reviewer’s valuable comments.

The original text is revised as follows:

  1. a)“The sillimanite was sourced from Guangdong Changlong Porcelain Company Ltd (Meizhou, China), and its median diameter was 32.15 μm (provided by the supplier).”
  2. b)“...and the 2 mL chloroform was added to each bottle to prevent microbial degradation of the organic acids and retard microbial growth.”
  3. c) “After the reaction for time periods of 24 h, 48 h, 72 h, 96 h, 120 h, 480 h, 360 h, 480 h, 600 h, and 720 h...”
  4. d) “..the mixtures were stirred for 3 min at the speed of 500rpm/min...”

Comment 3:

3.The Authors repeatedly describe in the manuscript the course of the dissolution curves as “The rate of increase in dissolution concentration was observed to be higher during the initial reaction time, followed by a plateau phase, and finally a decrease” (line 22, lines 135-139, lines 245-247). Meanwhile, the curves shown in Figure 2 and Figure 3 do not reflect such a course – it is difficult to observe a plateau, much less a decrease. The Authors should definitely comment/correct this.

Our response:

Thanks a lot for the valuable suggestions. The sentence has been corrected in the manuscript.

The original text is revised as follows:

“The rate of the dissolution concentration was observed to be higher during the initial reaction time, followed by slow down, and finally accelerated decrease. ”

Comment 4:

  1. 4. The greatest concern is the lack of data regarding the repeatability of the analyses performed for Al and Si concentrations. Are the results based on a single study, or did the Authors perform parallel measurements for several samples/solutions? What was the measurement error or standard deviation? In this context, does it make sense to discuss the differentvalues of released Al and Si ions for oxalic acid and citric acid? Are the differences obtained really statistically significant or are they within the range of the measurement error?

The Authors should comment on this and add appropriate information in the manuscript.

Our response:

Thanks a lot for the reviewer’s valuable comments.

Parallel measurements for the solutions was performed. The data of Al and Si concentrations used in the manuscript was the average value. The data is credible and the different values of released Al and Si ions for oxalic acid and citric acid implied the different dissolution mechanisms.

The original text is revised as follows:

“The liquid phase was stored in triplicate for later measurement, and the solid phase was placed in an 80 °C vacuum drying oven for 48 h to remove moisture and then stored for later use.”

“The Si concentrations were determined using the Si molybdenum blue spectrophotometric method with a visible light spectrophotometer (722N, Yoke Instrument Co., LTD, Shanghai, China). Parallel measurements for the solutions was performed, and the final value was averged.”

Comment 5:

  1. The results shown in Figure 4 should be discussed more thoroughly – the Authors limited themselves to only two lines of text (lines 148-149), meanwhile, the graphs show data that, in my opinion, require additional commentary, e.g. why are the values obtained for oxalic acid the highest, when the most Al and Si ions are released for a mixture of acids; what could be the reason for the sudden "jumps" of data in the course of the curves?

Our response:

Thanks a lot for the reviewer’s valuable comments.

The data in Figure 4 was calculated by the data (Si or Al concentration) in the Figure 2 and Figure 3. The sudden "jumps" suggested that the reduction of Si concentration or the increase of Al concentration for sillimanite in oxalic acid solution after reaction for 360 h. The reason for the sudden "jumps" of data in the course of the curves need futher investigation.

The original manuscript changes as follows:

“It was noteworthy that the Al/Si ratio in solution with organic acids was greater than 2 (Figure 4), proving that Al was easier to dissolve than Si in sillimanite (Al2SiO5). The results implied that the oxalic acid or citric acid had preferred dissolution of Al or Si. The values of the Al/Si ratio obtained for oxalic acid was the highest, as the Al was more easily extracted from the sillimanite. In the mixture of acids, the citric acid extracted Si from the sillimanite easily, compared with oxalic acid. Thus, the Al/Si ratio obtained for composite acids was not the most, when the most Al and Si ions are released for a mixture of acids.”

Comment 6:

  1. In lines 184-185 the Authors state that“Table 4 shows the I(120)/I(210) ofsillimanite before and after treatment with various organic acids”. Meanwhile, in Table 4 there are no data for I(120)/I(210) of sillimanite before acid treatment.

Our response:

Thanks a lot for the reviewer’s valuable comments. The sentence has been checked in the manuscript.

The original text is revised as follows:

“Table 4 shows the I(120)/I(210) of sillimanite after treatment with various organic acids.”

Comment 7:

  1. In lines 184-185 the Authors write that "The results of the XRD suggested that oxalic acid and citric acid had outstanding solubility on the (210) and (120) planes, respectively". Is it really outstanding? In my opinion, the differences in intensities for XRD patterns are not that pronounced.

Our response:

Thanks a lot for the reviewer’s valuable comments. The original manuscript changes as follows:

“ The results of the XRD suggested that oxalic acid and citric acid had preferred dissolution orientation on the (210) and (120) planes, respectively. ”

Comment 8:

  1. How do the Authors explain the identical course of all curves shown in Figure 2 in relation tothe already published data of dissolution of Si ions from kaolinite

(https://doi.org/10.1016/j.clay.2020.105756)?

Our response:

Thanks a lot for the reviewer’s valuable comments.

The phenomenon is interesting and further study is needed. The similar dissolution curves may attributed to the similar chemical environment of Si ions ([SiO4] tetrahedron and Si-O-Al linkage) in the sillimanite and kaolinte.

Comment 9:

  1. The manuscript also requires thorough editing, including:
  2. a) removal of double spaces, unnecessary blank lines, and insertion of missing paragraph spacing;
  3. b) The footnote in Table 1 lacks a subscript;
  4. c) line 102 – title of Subsection 2.3. should start with a capital letter;
  5. d) line 173 – Subsection 3.2. – the title should be completed "Changes in the crystal structure of sillimanite after treatment" – what kind of treatment/treatment with what?
  6. e) Line 185 – one ":" should be removed;
  7. f) The formatting of the literature list should also be adjusted to meet the requirements of the journal.

Our response:

Thanks a lot for the reviewer’s valuable comments.The original manuscript has been checked carefully. Some The changes in original manuscript are as follows:

“LOI1 = Loss on ignition.”

2.3 Characterization of the sample

3.2 Changes in the crystal structure of sillimanite after treatment with different orgainc acids

“Table 4 shows the I(120)/I(210) of sillimanite after treatment with various organic acids. The order was as follows: oxalic acid> complex acids> citric acid.”

………………………………….end of response to the comments……………………………

Round 2

Reviewer 1 Report

it can be accepted.

Author Response

We thank the referee very much for the comments.

Reviewer 2 Report

The manuscript can be accepted after minor corrections need to be made. Please revise the 'line' in Figure 5. The missing line causes the Figure to look incomplete, and the numbers (10, 20, .... 80) seem unconnected. 

Minor editing of English language required.

Author Response

Comment 1:

The manuscript can be accepted after minor corrections need to be made. Please revise the 'line' in Figure 5. The missing line causes the Figure to look incomplete, and the numbers (10, 20, .... 80) seem unconnected. 

Our response:

We thank the referee very much for the comments.

The Figure 5 has been checked carefully.

The original text is revised as follows:

Figure 5. The XRD pattern of sillimanite before and after treatment.”

Reviewer 3 Report

The Authors replied to most of the comments. However, I believe that some aspects of the review were not adequately addressed. Therefore my recommendation is still major revision.

Additional comments and suggestions for Authors are included in the attached file.

Moderate editing of English language is required.

Author Response

Comment 1:

The course of the dissolution curves is still described as follows: “The dissolution rate increased first, then remained constant, finally decreased” (line 22), “The rate of the dissolution concentration was observed to be higher during the initial reaction time, followed by slow down, and finally accelerated decrease” (lines 149-151), and “The dissolution rate increased initially, then stabilized before decreasing” (lines 292-293). 

Despite my sincere intentions, I am still unable to observe the part where the curves shown in Figures 2-3 stabilize or remain constant. Also the decrease stage is difficult to observe. I believe that the description of the Si and Al ion release curves still needs modifications.

Our response:

We thank the referee very much for the comments.

We have checked the manuscript carefully. The increase rate of dissolution concentration or the dissolution rate (Figure 1-1 and 1-2) was calculated by the data of Si and Al concentrations. The dissolution rate of Si and Al were found to be similar, with an decrease over time.

                        (a)                                (b)         

Figure 1-1 The (a) dissolution concentration and (b) dissolution rate of Si

                        (a)                                (b)         

Figure 1-2 The (a) dissolution concentration and (b) dissolution rate of Al

The original text is revised as follows:

“The increase rate of dissolution concentration of was high during the initial reaction time, then slow down, finally accelerated reduction, in the presence of LMWOAs. ”(Abstract)

“The increase rate of the dissolution concentration was observed to be higher during the initial reaction time, followed by a slow down, and finally accelerated decrease, in the presence of LMWOAs.”(Results and discussion)

“The increase rate of dissolution concentration of was high during the initial reaction time, then slow down, finally accelerated reduction, in the presence of low molecular weight organic acids. ”(Conclution)

Comment 2:

I still stand my comment on the statistics: The greatest concern is the lack of data

regarding the repeatability of the analyses performed for Al and Si concentrations. Are the results based on a single study, or did the Authors perform parallel measurements for several samples/solutions? What was the measurement error or standard deviation? In this context, does it make sense to discuss the different values of released Al and Si ions for oxalic acid and citric acid? Are the differences obtained really statistically significant or are they within the range of the measurement error?

The Authors should comment on this and add appropriate information in the manuscript. In my opinion, it is still not clearly stated how many measurements were made for a given sample taken after a certain reaction time. Does the notation "The liquid phase was stored in triplicate for later measurement" mean that 3 replicates were made for each sample? If so, what was the standard deviation for these replications? Furthermore, the SD data should be included in the graphs shown in Fig. 2 and Fig. 3. By the way, the sentence introduced to the manuscript: “Parallel measurements for the solutions was performed, and the final value was averged.” (lines 133-134) needs thorough English correction.?

Our response:

Thanks a lot for the reviewer’s valuable comments. "The liquid phase was stored in triplicate for later measurement" means that 3 replicates were made for each sample. The standard deviation for these replications was included in the Figure 1 and Figure 2 in the manuscript.

“Parallel measurements for the solutions was performed, and the final value was averaged.” means that the general test was conducted, which is common. Thus, the sentence was deleted and the paragraph could also be understood.

Comment 3:

      r/min should be replaced by rpm/min.

Our response:

Thanks a lot for the valuable suggestions. The sentence has been corrected in the manuscript.

The original text is revised as follows:

“The solution was stirred at 500 rpm/min for 10 minutes, sealed, and placed in a biochemical incubator at 25°C. At regular intervals, the plastic bottles were moved and stirred at 500 rpm/min for 2 min.”

Comment 4:

      Moderate editing of English language is required.

Our response:

Thanks a lot for the valuable suggestions. The English language in the manuscript have been carefully checked. The original text is revised as follows:

“The dissolution concentrations of Si and Al in composite organic acids were more than that in single organic acid, with maximum values of 1754 μM and 3904 μM, respectively.”

“In the industry, the sillimanite is added to other products as a raw material.”

“The sillimanite’s crystal structure is similar with mullite. Generally, sillimanite decomposes to mullite and silica at high temperature, with a certain volumetric expansion.”

………………………………….end of response to the comments……………………………
